EMBO
Molecular Medicine

# SPINK2 deficiency causes infertility by inducing sperm defects in heterozygotes and azoospermia in homozygotes

Zine-Eddine Kherraf[1,†], Marie Christou-Kent[1,†], Thomas Karaouzène[1], Amir Amiri-Yekta[1,2,3], Guillaume Martinez[1], Alexandra S Vargas[1], Emeline Lambert[1], Christelle Borel[4], Béatrice Dorphin[5], Isabelle Aknin-Seifer[6], Michael J Mitchell[7], Catherine Metzler-Guillemain[7], Jessica Escoffier[1], Serge Nef[4], Mariane Grepillat[1], Nicolas Thierry-Mieg[8], Véronique Satre[1,9], Marc Bailly[10,11], Florence Boitrelle[10,11], Karin Pernet-Gallay[12], Sylviane Hennebicq[1,13], Julien Fauré[2,12], Serge P Bottari[1,14], Charles Coutton[1,9], Pierre F Ray[1,2,‡,*] 🆔 & Christophe Arnoult[1,‡]

## Abstract

Azoospermia, characterized by the absence of spermatozoa in the ejaculate, is a common cause of male infertility with a poorly characterized etiology. Exome sequencing analysis of two azoospermic brothers allowed the identification of a homozygous splice mutation in *SPINK2*, encoding a serine protease inhibitor believed to target acrosin, the main sperm acrosomal protease. In accord with these findings, we observed that homozygous *Spink2* KO male mice had azoospermia. Moreover, despite normal fertility, heterozygous male mice had a high rate of morphologically abnormal spermatozoa and a reduced sperm motility. Further analysis demonstrated that in the absence of Spink2, protease-induced stress initiates Golgi fragmentation and prevents acrosome biogenesis leading to spermatid differentiation arrest. We also observed a deleterious effect of acrosin overexpression in HEK cells, effect that was alleviated by SPINK2 coexpression confirming its role as acrosin inhibitor. These results demonstrate that SPINK2 is necessary to neutralize proteases during their cellular transit toward the acrosome and that its deficiency induces a pathological continuum ranging from oligoasthenoteratozoospermia in heterozygotes to azoospermia in homozygotes.

**Keywords** azoospermia; exome sequencing; genetics; infertility; spermatogenesis

**Subject Categories** Genetics, Gene Therapy & Genetic Disease; Urogenital System

## Introduction

The World Health Organization estimates that 50 million couples worldwide are confronted with infertility. Assisted reproduction technologies (ART) initiated 35 years ago by Nobel Prize Winner Robert Edwards have revolutionized the practice of reproductive medicine, and it is now estimated that approximately 15% of couples in Western countries seek assistance from reproductive clinics for infertility or subfertility. Despite technological breakthroughs and advances, approximately half of the couples concerned still fail to achieve a successful pregnancy even after repeated treatment cycles. Alternative treatment strategies should therefore be

1 Genetic Epigenetic and Therapies of Infertility, Institute for Advanced Biosciences, Inserm U1209, CNRS UMR 5309, Université Grenoble Alpes, Grenoble, France
2 CHU de Grenoble, UF de Biochimie Génétique et Moléculaire, Grenoble, France
3 Department of Genetics, Reproductive Biomedicine Research Center, Royan Institute for Reproductive Biomedicine, ACECR, Tehran, Iran
4 Department of Genetic Medicine and Development, University of Geneva Medical School, Geneva 4, Switzerland
5 Laboratoire d'Aide Médicale à la Procréation, Centre AMP 74, Contamine-sur-Arve, France
6 Laboratoire de Biologie de la Reproduction, Hôpital Nord, Saint Etienne, France
7 Aix Marseille Univ, INSERM, GMGF, Marseille, France
8 Univ. Grenoble Alpes / CNRS, TIMC-IMAG, Grenoble, France
9 CHU de Grenoble, UF de Génétique Chromosomique, Grenoble, France
10 Department of Reproductive Biology and Gynaecology, Poissy General Hospital, Poissy, France
11 EA 7404 GIG, Université de Versailles Saint Quentin, Montigny le Bretonneux, France
12 Grenoble Neuroscience Institute, INSERM 1216, Grenoble, France
13 CHU de Grenoble, UF de Biologie de la procréation, Grenoble, France
14 CHU de Grenoble, UF de Radioanalyses, Grenoble, France
*Corresponding author. Tel: +33 4 76 76 55 73; E-mail: pray@chu-grenoble.fr
†These authors contributed equally to this work
‡These authors contributed equally to this work as senior authors

envisaged to improve ART success rate, especially for patients impervious to usual assisted reproductive technologies. Improvement in treatment efficiency essentially depends upon an accurate diagnosis and the characterization of the molecular etiology of the defect. These efforts to better characterize infertility subtypes should first be concentrated on the most severe defects since they generally have a poor prognosis and affected patients would benefit the most from new treatments. Moreover, the most severe phenotypes are more likely to be caused by monogenic defects which are easier to identify. As such, the genetic exploration of non-obstructive azoospermia (NOA), the absence of spermatozoa in the ejaculate due to a defect in spermatogenesis, should be considered a priority. NOA is a common cause of infertility found in approximately 10% of the couples assessed for infertility. Although a genetic etiology is likely to be present in most cases of azoospermia, only a few defective genes have so far been associated with this pathology accounting for a minority of cases. At present, only chromosomal abnormalities (mainly 47XXY, Klinefelter syndrome identified in 14% of cases) and microdeletions of the Y chromosome are routinely diagnosed, resulting in a positive genetic diagnosis in < 20% of azoospermia cases (Tuttelmann *et al*, 2011). The evolution of sequencing technologies and the use of whole-exome or whole-genome sequence (WES/WGS) analysis paves the way to a great improvement in our ability to characterize the causes of genetically heterogeneous pathologies such as NOA.

Spermatogenesis can be subdivided into three main steps: (i) multiplication of diploid germ cells; (ii) meiosis, with the shuffling of parental genes and production of haploid cells; and (iii) spermiogenesis, the conversion of round spermatids into one of the smallest and most specialized cells in the body, the spermatozoa. NOA is expected to be mainly caused by failures in steps 1 and 2, and it is indeed what has been observed in a majority of cases so far. Very recently, defects in six genes were linked to azoospermia in man. Most of these genes code for meiosis-controlling proteins such as TEX11, TEX15, SYCE1, or MCM8, and the absence of the functional proteins induces a blockage of meiosis (Tuttelmann *et al*, 2011; Maor-Sagie *et al*, 2015; Okutman *et al*, 2015; Yang *et al*, 2015; Yatsenko *et al*, 2015). Another WES analysis of two consanguineous families identified likely causal mutations in *TAF4B* and *ZMYND15* (Ayhan *et al*, 2014). Study of *Taf4b* KO mice showed that homozygous mutant males are subfertile with extensive pre-meiotic germ cell loss due to altered differentiation and self-renewal of the spermatogonial stem cell pool, thus illustrating that pre-meiotic block induces NOA. More surprisingly, *ZMYND15* codes for a spermatid-specific histone deacetylase-dependent transcriptional repressor and its absence in mice induced a significant depletion of late-stage spermatids (Yan *et al*, 2010) suggesting that NOA can also be induced by post-meiotic defects.

Here, WES analysis of two brothers with NOA led to the identification of a homozygous truncating mutation in the *SPINK2* gene coding for a Kazal family serine protease inhibitor. Studying KO mice, we observed that homozygous KO animals also suffered from azoospermia thus confirming the implication of SPINK2 in NOA. Furthermore, we observed that SPINK2 is expressed from the round-spermatid stage onwards thus confirming that post-meiotic anomalies can result in NOA. We suggest that SPINK2 is necessary to neutralize the action of acrosomal proteases shortly after their synthesis and before they can be safely stored in the acrosome where they normally remain dormant until their release during the acrosome reaction. We also show that in the absence of SPINK2, protease-induced stress initiates Golgi fragmentation contributing to the arrest of spermatid differentiation and their shedding from the seminiferous epithelium. The characterization of the molecular pathophysiology of this defect opens several novel therapeutic perspectives which may allow the restoration of a functional spermatogenesis.

# Results

## Medical assessment of two brothers with defective sperm production

Two French brothers (Br1 and Br2), born from second cousin parents (Fig 1A), and their respective wives sought medical advice from infertility clinics in France (Chatellerault, Tours, Poissy, and Grenoble) between 2005 and 2014 after 2 years of unsuccessful attempts to spontaneously conceive. Analyses of their ejaculates (Fig 1B; $n = 5$) evidenced the absence of spermatozoa for the first brother (Br1) and a very low concentration (0–200,000/ml, mean 126,000/ml $n = 5$) for the second (Br2). Moreover, all spermatozoa were immotile and presented an abnormal morphology (pin-shaped head devoid of acrosome; detached flagella) and were not suitable for *in vitro* fertilization (IVF) with intracytoplasmic sperm injection (ICSI). Interestingly, ejaculates of both brothers presented a significant concentration of germ cells ($8.6 \times 10^6 \pm 6.2 \times 10^6$/ml and $9.0 \times 10^6 \pm 7.0 \times 10^6$/ml for Br-1 and Br-2, respectively) likely corresponding to spermatids. As Br1 and Br2 both present a severe default of sperm production with a high number of spermatids in the ejaculate, we believe that they present the same phenotype, likely caused by the same genetic defect. A normal karyotype was observed for both brothers (46,XY), and no deletions of the Y

**Figure 1. Azoospermia in two consanguineous brothers.**

A    Genetic tree of the studied family showing affected brothers Br1 and Br2 illustrating the consanguinity of the parents (P1 and P2).

B    Comparisons of ejaculate volume ($n = 5$) and spermograms ($n = 5$) of brothers Br1 and Br2 with those of fertile controls ($n = 35$) evidence the absence of mature sperm and the presence of round cells in the ejaculates. Data represent mean ± SEM. *P*-values are $P = 4 \times 10^{-4}$ (a), $P = 0.6$ (b, non-significant), and $P = 4 \times 10^{-5}$ (c); statistical differences were assessed using *t*-test.

C, D    Testis sections from a fertile control and (D) patient Br1 stained with periodic acid–Schiff (PAS). The lumen of tubules from the control is large and mature sperm are present (C), whereas the lumen of most of seminiferous tubules from patient Br1 is filled with non-condensed and early condensed round spermatids and no mature sperm are observed. Scales bars, 100 μm.

E, F    In the fertile control (E) seminiferous tubule cross sections, spermatogonia (Sg), spermatocytes (Sc) and spermatids (RS) are regularly layered, whereas the different types of spermatogenic cells are disorganized in patient Br1 (F). Scales bars, 100 μm.

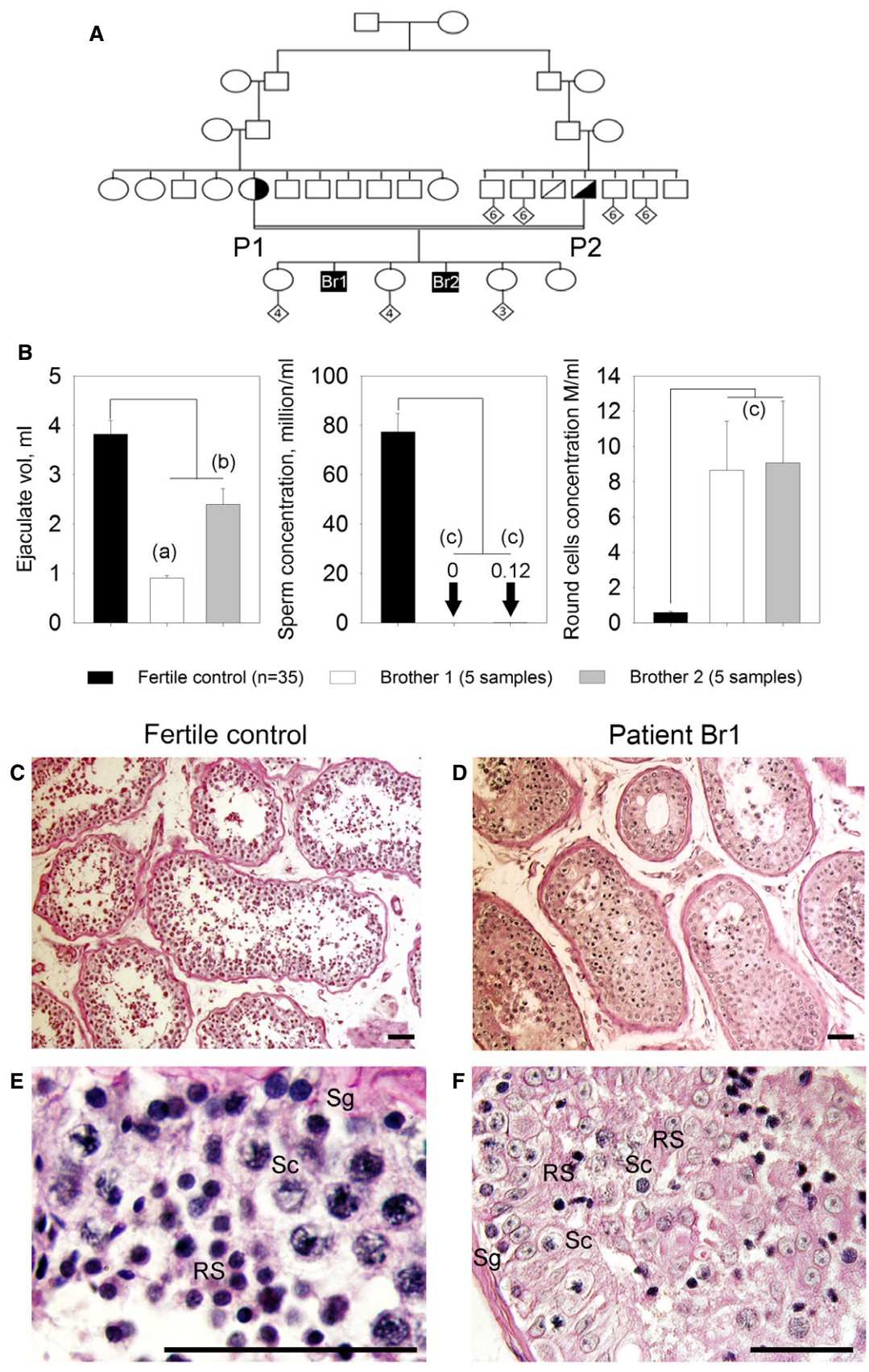

**Figure 1.**

chromosome were observed at the AZF loci. Testis sperm extraction was carried out twice for Br1 in 2008 and 2014. Each time the recovery was unsuccessful (although a few spermatozoa were observed in fixed dilacerated testicular tissues) suggesting a diagnosis of post-meiotic NOA. Histological analysis of seminiferous tubules obtained from Br1 biopsies showed: (i) a disorganization of the structure of the tubules; (ii) that the lumen of the seminiferous tubules were filled with immature germ cells, an indication of intense desquamation of the germinal epithelium; and (iii) a reduced number of round spermatids, with an overrepresentation of early round spermatids (Fig 1C–F). Brother Br2 has only had spermograms for diagnostic purposes which did not show any ICSI-compatible spermatozoa and has not been able to attempt ART.

### Whole-exome sequencing identifies a homozygous truncating mutation in *SPINK2*

Since the brothers were married to unrelated women, we excluded the possibility of a contributing female factor and focused our research on the brothers. Given the familial history of consanguinity, we postulated that their infertility was likely caused by a common homozygous mutation. We proceeded with WES to identify a possible genetic defect(s) which could explain the observed azoospermia. After exclusion of common variants, both bothers carried a total of 121 identical missense heterozygous variants (none appearing as obvious candidate) and only five identical homozygous variants common to both brothers (Appendix Table S1). Among these different genes, only the Chr4:57686748G>C *SPINK2* variant was described to be predominantly expressed in human testis (Appendix Fig S1A) as well as in mouse testis (Appendix Fig S1B). The mutation was validated by Sanger sequencing in both brothers (homozygous) and their parents (heterozygous) (Fig 2A). *SPINK2* thus appeared as the best candidate to explain the human condition. The variant Chr4:57686748G>C was not present in > 121,000 alleles analyzed in the ExAC database (http://exac.broadinstitute.org) and could have an effect on RNA splicing. *SPINK2* is located on chromosome 4 and contains four exons (Fig 2B). The gene codes for a Kazal type 2 serine protease inhibitor also known as an acrosin–trypsin inhibitor. The Ensembl expression database (www.ensembl.org) predicts the presence of four transcripts. We studied the expression of the different transcripts in human testis by RT–PCR, and only one band was present corresponding to NM_021114, ENST00000248701, which codes for a protein of 9.291 kDa consisting of 84 amino acids (Fig 2C). All nucleotide sequences herein refer to this transcript. The identified mutation, c.56-3C>G, is located three nucleotides before exon 2 and may create a new splice acceptor site, leading to a frameshift and premature stop codon in exon 2 and the generation of an abnormal transcript (T1) and/or to the skipping of exon 2 (44 nt) giving rise to an early stop codon at the beginning of exon 3 and the generation of another abnormal transcript (T2) (Fig 2B). To validate these hypotheses, RT–PCR was performed on testicular extract from Br1. Two bands were observed (Fig 2C) and sequenced after isolation of each band following gel electrophoresis. Sequence analysis demonstrated that the bands corresponded to T1 and T2, demonstrating that both abnormal transcripts were present in the patient's testis (Fig 2D). Since the protease inhibitor and binding sites of the protein are coded mostly by exon 3, it is expected that the truncated proteins corresponding to T1 and T2 transcripts are not functional

(Appendix Fig S2). Sequencing of Br1's transcripts therefore confirms that the identified splice variant abrogates the production of a full-length protein thereby confirming its role as a deleterious mutation.

### Importance of *SPINK2* variants as a cause for human infertility: sequence analysis of a cohort of infertile men with an altered spermatogenesis

We sequenced *SPINK2* whole coding sequences of 611 patients affected by azoo- or oligozoospermia (210 patients with azoospermia, 393 subjects with oligozoospermia and 8 with unspecified cause). Only one variant, identified in patient 105 (P105), was not described in ExAC and was likely deleterious (Appendix Table S2). This variant, c.1A>T (Fig EV1A), abrogates the *SPINK2* start codon and was present heterozygously in P105, a man with oligozoospermia. An alternate start site could potentially be used in the middle of exon 3 allowing the synthesis of a truncated protein of 2 kDa lacking the reactive site and disulfide bonds, both known to be crucial for SPINK2 function (Fig EV1B). However, overexpression of the mutated gene in HEK cells did not produce any portion of the SPINK2 protein indicating that the putative alternative start site is not functional and that the alteration of the initial start site does not permit the synthesis of any part of the SPINK2 protein. This was evidenced by transfecting HEK293 cells with a plasmid containing the full human *SPINK2* ORF sequence with the c.1A>T mutation and a C-terminus DDK-tagged. Extracted proteins were loaded onto a 20% acrylamide gel and detected with anti-DDK or anti-SPINK2 antibodies (Fig EV1C). P105 and his wife, born from non-consanguineous parents, experienced a 5-year period of infertility before giving birth to a healthy boy conceived spontaneously. They sought medical advice 2 years after their son's birth to initiate a second pregnancy. Sperm analysis resulted in the diagnosis of oligozoospermia associated with a reduced percentage of progressive motile spermatozoa (Table EV1). The patient's sperm morphology was assessed with Harris–Shorr staining using the modified David's classification and showed that 34–39% of sperm had a normal morphology (*n* = 2). The main defects observed were abnormal acrosome (34–39%) and defective neck–head junction (40–46%), defects that are similar to those observed in patient Br2.

This analysis indicates that *SPINK2* defects are extremely rare with an allelic frequency of approximately 1/1,200 in the cohort of infertile men analyzed. The rarity of *SPINK2* variants and the fact that P2, the father of Br1 and Br2, also harboring a heterozygous mutation, presents in a milder phenotype than P105 could indicate that *SPINK2* haploinsufficiency induces a milder phenotype of oligozoospermia with an incomplete penetrance on infertility.

### Homozygous *Spink2* KO mice have azoospermia due to a spermiogenesis blockade at the round-spermatid stage

In order to confirm that the absence of SPINK2 leads to azoospermia, homozygous *Spink2* KO ($^{-/-}$) mice were obtained and their reproductive phenotype was studied. We first performed qRT–PCR on *Spink2*$^{+/+}$ and *Spink2*$^{-/-}$ testis mRNA extracts to validate the absence of *Spink2* mRNA and thus of protein. Contrary to what was observed in WT littermates, we observed no *Spink2* amplification in KO males, confirming Spink2 deficiency (Appendix Fig S3). Males

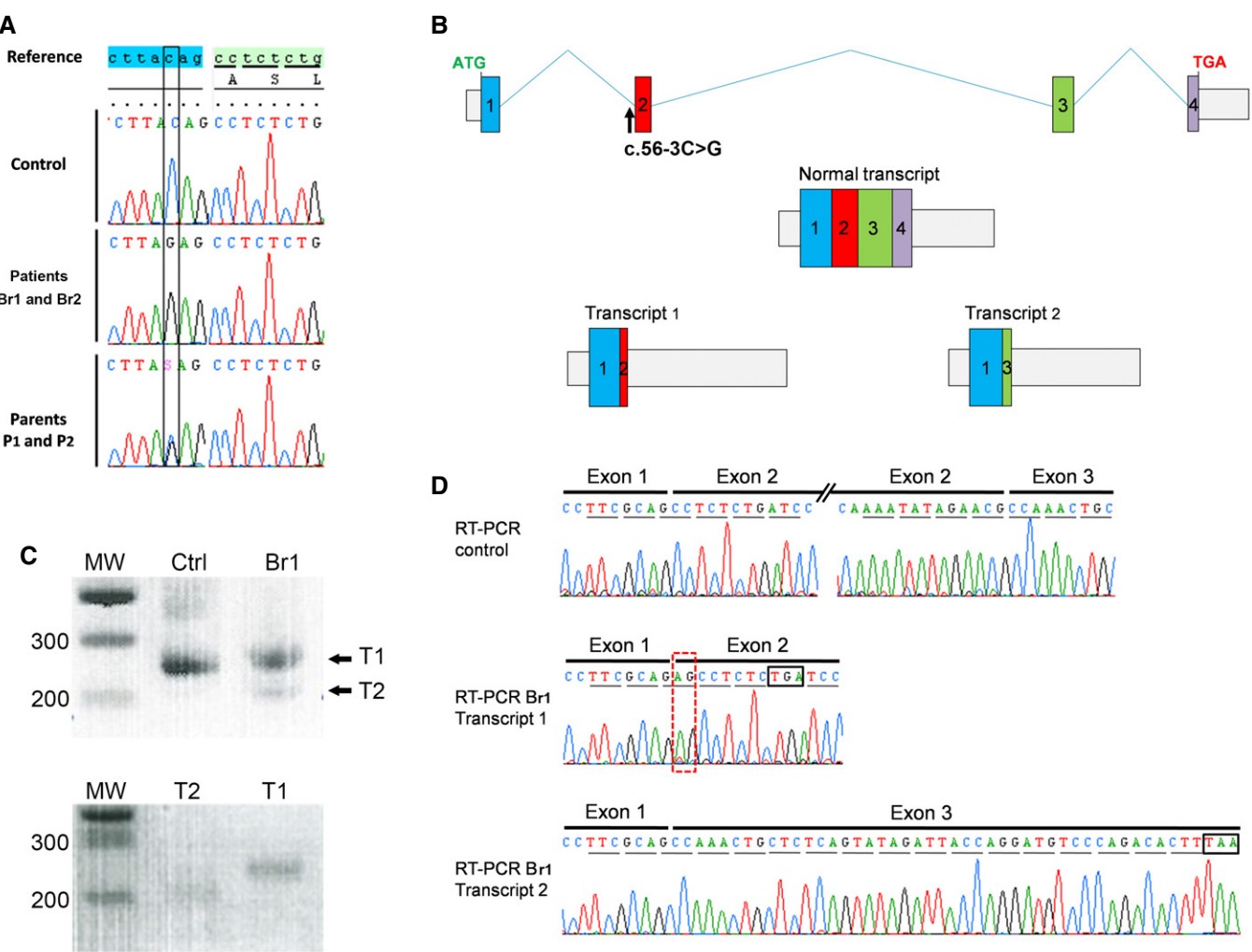

**Figure 2. Identification of a *SPINK2* variant (c.56-3C>G) by exome sequencing and its consequences on splicing and translation.**

A   The identified variant, homozygous in patients 1 and 2 and heterozygous in their parents, is located three nucleotides before exon 2 and creates an AG that immediately precedes the original AG splice acceptor site.

B   If recognized during splicing, this new acceptor site is expected to add two nucleotides (AG) at the beginning of exon 2, inducing a frameshift leading to a stop codon 3 amino acids later (transcript 1). The non-recognition of the abnormal acceptor site is expected to induce the skipping of exon 2 (transcript 2). The first stop codon can be observed 15 codons after the mis-inserted exon 3.

C   RT–PCR of mRNA extracts from fertile control (Ctrl) and the brother Br1. Results show one band for Ctrl. The sequencing of this band showed that it corresponds to transcript NM_021114. For Br1, two bands were present, named T1 and T2. Bottom gel shows T1 and T2 after gel isolation.

D   Transcripts T1 and T2 were collected and sequenced: T1 showed the insertion of an additional AG (red-dashed rectangle) leading to a premature stop codon (black box), whereas transcript T2 showed that exon 2 had been excised; these two transcripts correspond to the expected transcripts 1 and 2 from panel (B). Stop codons are shown in black boxes.

were completely infertile, whereas no reproductive defects were observed in females (Fig 3A1). Homozygous KO mice had comparatively smaller sized testes and a testis/body weight ratio half of their wild-type (WT) littermates [$3.63 \pm 0.21$ in WT and $1.77 \pm 0.03$ in KO (Fig 3A2)]. Furthermore, there was a complete absence of spermatozoa in *Spink2*$^{-/-}$ caudal epididymis (Fig 3A3) which only contained round cells likely corresponding to round spermatids and multinucleated cells, known as symblasts. Histological studies of KO seminiferous tubules stained with periodic acid–Schiff (PAS) revealed the presence of germ cells up to the early round-spermatid stage but condensed and elongated spermatids and

mature spermatozoa were completely absent, contrary to WT (Fig 3B1 and C1). The lumen of the seminiferous tubules of *Spink2*$^{-/-}$ males contained round cells and symblasts (Fig EV2A and B), a result in agreement with observations of the cellular content of the cauda epididymis, which showed the presence of round cells only (Fig EV2C). In contrast to what was observed in WT (Fig 3B2), sections of caudal epididymis confirmed the absence of spermatozoa and the presence of symblasts and round cells (Fig 3C2). Comparing PAS staining of *Spink2*$^{+/+}$ and *Spink2*$^{-/-}$ seminiferous tubules, we noticed that contrary to WT, *Spink2*$^{-/-}$ round spermatids did not contain an acrosomal vesicle, suggesting

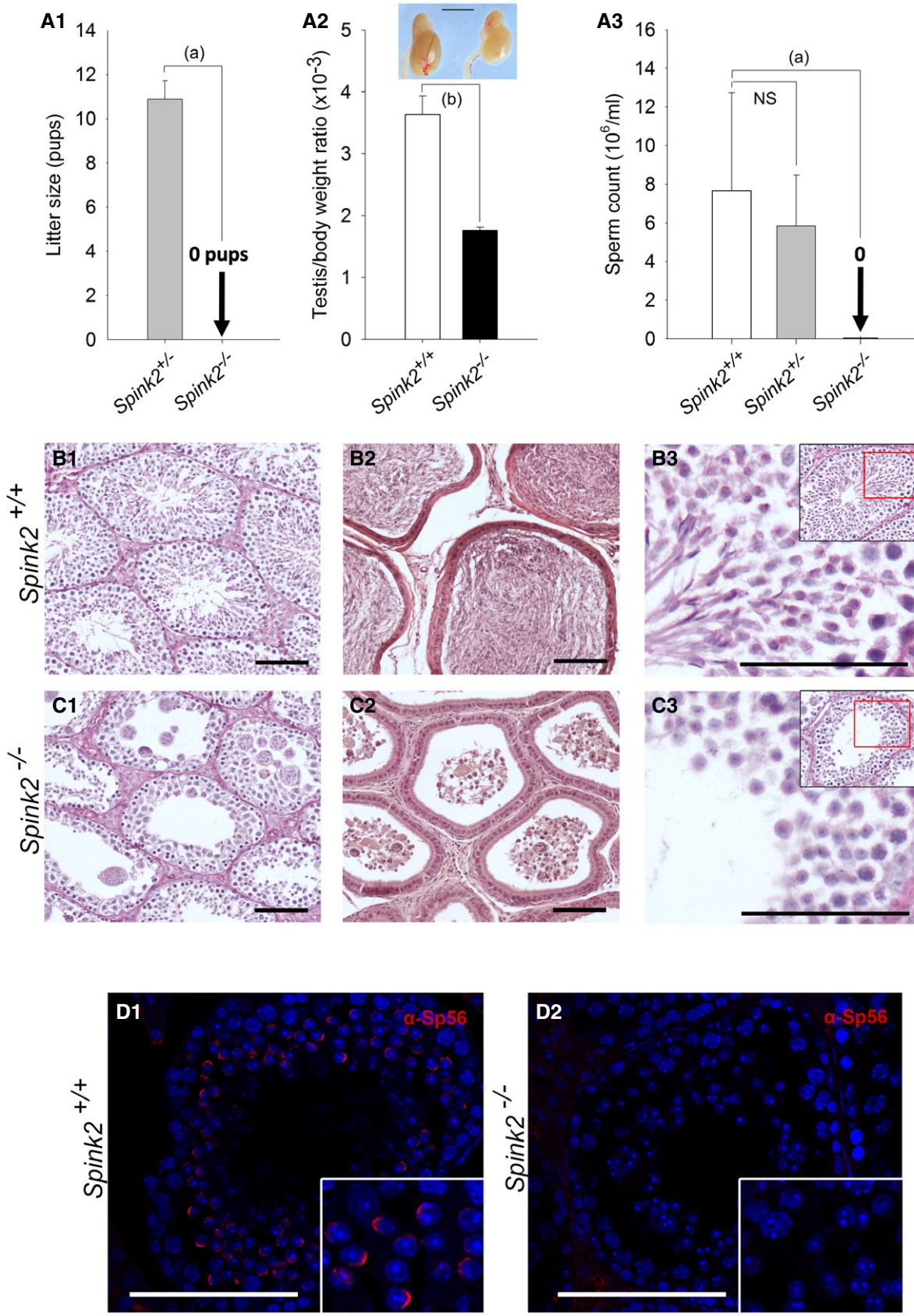

**Figure 3.**

◀

**Figure 3.**  *Spink2⁻/⁻* males are infertile and azoospermic, and spermatogenesis presents a post-meiotic blockade.

A1    Litter size of *Spink2⁻/⁻* and *Spink2⁺/⁻* males mated with wild-type females (*n* = 5).
A2    Testis/body weight ratio for WT and *Spink2⁻/⁻* mice (*n* = 6) and morphology and size of wild-type and *Spink2⁻/⁻* testes of male siblings. Scale bar, 5 mm.
A3    Sperm concentrations from the cauda epididymis of wild-type, *Spink2⁺/⁻*, and *Spink2⁻/⁻* male testes (*n* = 10).
B, C   Histological comparisons of testis and epididymis from WT and *Spink2⁻/⁻* mice. (B1, C1) Periodic acid–Schiff (PAS) staining of seminiferous tubule cross sections shows complete spermatogenesis in WT (B1) contrary to *Spink2⁻/⁻* mice (C1), where condensed, elongated spermatids and mature sperm are absent. (B2, C2) Sections of epididymis stained with eosin/hematoxylin. In the lumen of tubules from WT mice, mature sperm are present (B2), whereas only round cells and multinucleated symblasts occupy the lumen of tubules from *Spink2⁻/⁻* mice (C2). (B3, C3) Enlargement of seminiferous tubule sections stained with PAS evidences deep pink staining in round spermatids, which corresponds to the acrosome in WT mice (B3), whereas round spermatids from *Spink2⁻/⁻* mice present no deep pink staining, indicating that the acrosome is not formed (C3). Scale bars, 100 μm.
D1, D2   Immunofluorescence experiments using an anti-Sp56 antibody (red staining) confirm the presence of the acrosome in seminiferous tubule sections from WT contrary to those from *Spink2⁻/⁻* mice, where no staining is observed. Scale bars, 100 μm.

Data information: Data represent mean ± SEM. *P*-values are $P = 1 \times 10^{-5}$ (a) and $P = 1 \times 10^{-4}$ (b); statistical differences were assessed using *t*-test. NS, not statistically significant.

that the absence of Spink2 prevents acrosome biogenesis (Fig 3B3 and C3). This point was confirmed by immunofluorescent staining using the Sp56 antibody, a specific marker of the acrosome (Kim *et al*, 2001) (Fig 3D1 and D2). We then identified the spermatogonia using an anti-PLZF antibody (Zhang *et al*, 2014) (Fig EV3A and B) and observed no significant difference in the median number of spermatogonia per tubule (*n* = 30) between *Spink2⁺/⁺* and *Spink2⁻/⁻* mice (Fig EV3C). These results indicate that the absence of Spink2 does not impact spermatogonial survival but leads to an early arrest of round-spermatid differentiation. Overall, the *Spink2⁻/⁻* mouse phenotype perfectly mimics the human condition and confirms that SPINK2 deficiency is involved in human azoospermia.

**SPINK2 is an acrosomal protein**

In order to further investigate the molecular pathogeny of this SPINK2-dependent azoospermia, we determined the localization of SPINK2 in human and mouse testis. We first verified the specificity of a SPINK2 antibody through Western blot (WB) and immunofluorescence (IF) experiments on HEK293 cells overexpressing human SPINK2. In Western blots, the SPINK2 antibody recognized three bands of less than 17 kDa weight, likely corresponding to oligomeric complexes (Appendix Fig S4A). No bands appeared in nontransfected cells. Moreover, the overexpressed SPINK2 featured a DDK-tag which was recognized by an anti-DDK-tag antibody revealing three bands of identical molecular mass (Appendix Fig S4B). No bands were observed when the primary antibody was omitted. SPINK2 expression was also studied by IF and confocal microscopy. Transfected cells displayed a cytoplasmic staining, whereas no staining was observed in non-transfected cells (Appendix Fig S4C). Taken together, these results demonstrate the specificity of this antibody in WB and IF experiments. Next, the localization of SPINK2 was determined by IF in human and mouse seminiferous tubule cross sections and in mature sperm (Fig EV4). In mouse, SPINK2 was present in the acrosomal vesicle from the beginning of the acrosome's biogenesis at the round-spermatid stage as indicated by a colocalization with Sp56, a marker of the acrosome (Fig EV4A and B). In accordance with the results shown in Fig 3D2, no SPINK2 staining was observed in *Spink2⁻/⁻* testis cross sections (Fig EV4C). A similar localization was observed for SPINK2 in human seminiferous tubule sections (Fig EV4D). Finally, we observed that SPINK2 remains present in the acrosome of human and mouse mature spermatozoa (Fig EV4E and F).

**Ultrastructure of *Spink2⁻/⁻* round spermatids shows that fusion of proacrosomal vesicles is hampered and that the Golgi apparatus is fragmented**

We showed that SPINK2 is located in the acrosome and that its absence prevents acrosome biogenesis. To understand the reasons for the absence of acrosome biogenesis, we performed transmission electronic microscopy (EM) to study the ultrastructure of round spermatids from *Spink2⁻/⁻* males (Fig 4). In wild-type round spermatids, proacrosomal vesicles generated by the Golgi apparatus docked in a specialized area of the nuclear envelope (NE) and fused together to form a giant acrosomal vesicle (Fig 4A). Contrary to WT, in *Spink2⁻/⁻*, the proacrosomal vesicles generated by the Golgi apparatus of round spermatids were mostly unable to fuse (Fig 4B2, white arrowheads), likely explaining the absence of acrosome biogenesis. Moreover, the Golgi apparatus from *Spink2⁻/⁻* animals produced abnormal proacrosomal vesicles of irregular sizes (Fig 4B2) and showed a considerable disorganization with a decreased proportion of flattened membrane stacks (Fig 4B2) displaying shorter lengths (Fig 4C). Acrosome biogenesis is dependent on the simultaneous synthesis of vesicles by the Golgi apparatus and the modification of the nuclear envelope (NE) facing the Golgi apparatus, with tight apposition of both nuclear membranes and aggregation of a nuclear dense lamina (NDL) on the nuclear side of the inner nuclear membrane (Kierszenbaum *et al*, 2003). In *Spink2⁻/⁻* round spermatids, the densification of the NE appears to occur normally and the NDL is clearly visible in EM (Fig 4B2). Using IF, the modification of the NE facing the Golgi apparatus was followed with an anti-Dpy19l2 antibody. We indeed had previously shown that Dpy19l2 participates in linking the acrosome to the nucleus and that it is located in the nuclear membrane facing the forming acrosome (Pierre *et al*, 2012) and is thus a component of this specialized area of the nuclear envelope. In costaining experiments using anti-Dpy19l2 and anti-GM130 antibodies to stain the nuclear envelope facing the acrosomal vesicle (evidenced by the NDL in EM) and the Golgi apparatus, respectively, we found that in WT round spermatids, the Golgi apparatus is either located immediately in front of the NDL in the early phase of acrosome biogenesis or, at a slightly later stage, lies adjacent to it (Fig 4D1 and D2). In contrast to WT, the Golgi apparatus of *Spink2⁻/⁻* round spermatids was positioned randomly around the nucleus, often found on the opposite side of the NDL (Fig 4D3–D6) indicative of a disruption of the polarity of the NDL and of the Golgi apparatus, which should both be located at the apical face of the round spermatid.

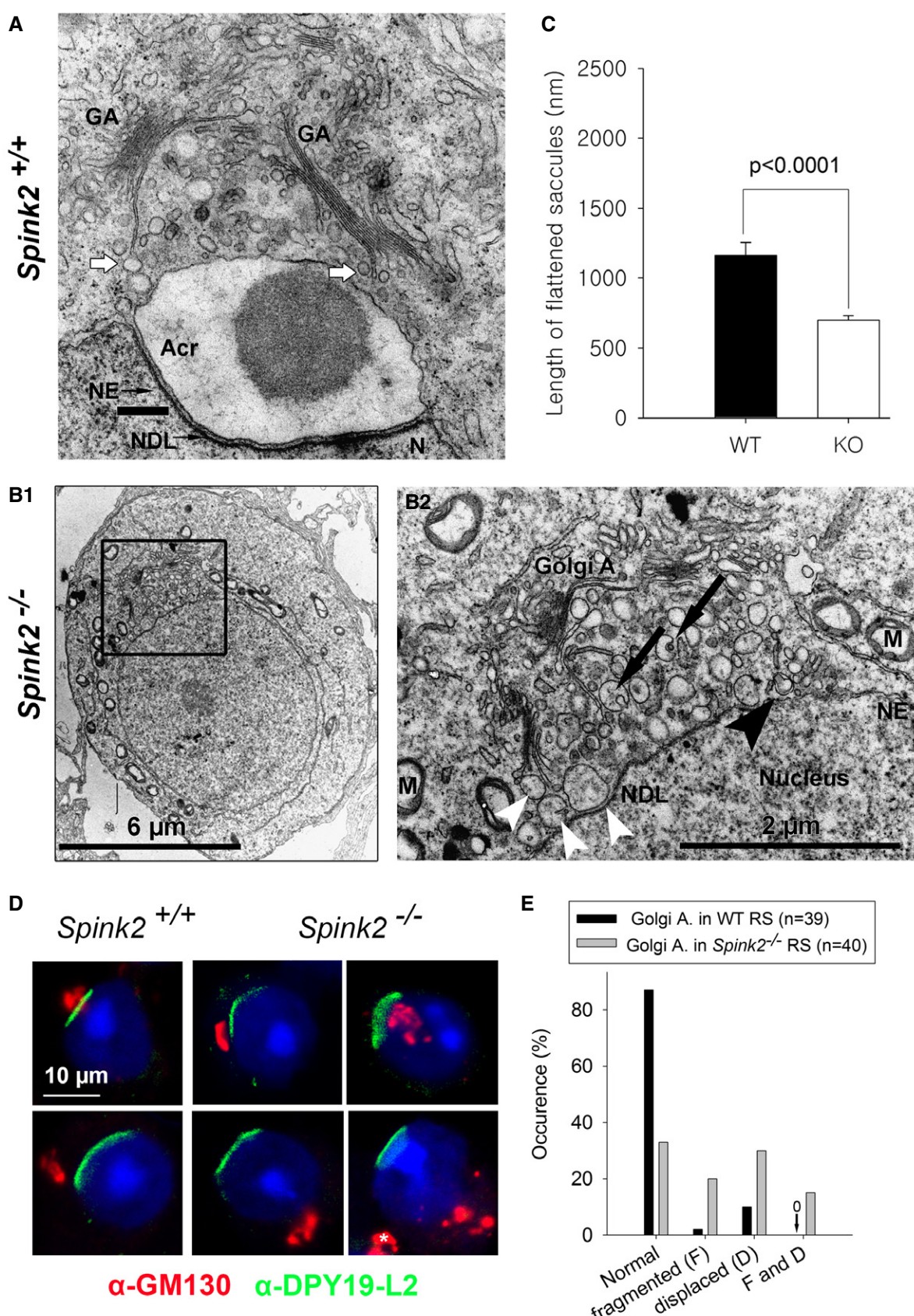

**Figure 4.**

◄

**Figure 4.  Lack of Spink2 prevents the fusion of proacrosomal vesicles and induces a disorganization of the Golgi apparatus.**

A  Partial section of a WT round spermatid observed by EM showing the early biogenesis of the acrosome (Acr) due to the continuous formation and aggregation of small vesicles (white arrows) coming from the Golgi apparatus (GA). The nuclear envelope (NE) facing the acrosome has a specific organization and is associated with the nuclear dense lamina (NDL). N, nucleus. Scale bar, 400 nm.

B  Ultrastructure of the Golgi apparatus in *Spink2$^{-/-}$* round spermatid observed by EM. (B1) Ultrastructure of a *Spink2$^{-/-}$* round spermatid observed at low magnification. The black box corresponds to the Golgi apparatus and is enlarged in (B2). (B2) In the absence of Spink2, vesicles do not aggregate at the nuclear envelope although modification of the NE and formation of the NDL occur. Unfused vesicles of different sizes accumulate in the cytoplasm with very few docking on the nuclear envelope (white arrowhead). Moreover, the GA shows disorganization with strong decrease or absence of stacks of flattened membranes. Finally, microautophagy-like structures and vesicles with a double membrane (black arrowhead) are observed around the GA (black arrows). M, mitochondria. Scale bars, 6 μm (B1) and 2 μm (B2).

C  The length of flattened saccules is statistically reduced in *Spink2$^{-/-}$* round spermatids (WT saccules, *n* = 74; and KO saccules, *n* = 136). Data represent mean ± SEM; the statistical difference was assessed with *t*-test, *P*-value as indicated.

D  Absence of Spink2 induces Golgi apparatus fragmentation and mislocalization. (D1, D2) IF experiments using an anti-Dpy19l2 antibody marking the specific NE facing the NDL (green staining) and an anti-GM130 antibody marking the cis-Golgi (red staining) show that the Golgi apparatus (GA) is a compact structure and located either in front of the NDL or close to it in WT round spermatids (normal). (D3–D6) In contrast, similar double staining of round spermatids from *Spink2$^{-/-}$* mice shows that only one-third of GA are compact and normally placed (D3) and the other GA are either displaced (D4), fragmented (D5), or both (D6). In panel (D6), white asterisk corresponds to a GA belonging to a different cell.

E  Quantification of the morphology and the relative localization of the GA and Dpy19l2 staining in WT (*n* = 40) and *Spink2$^{-/-}$* (*n* = 39) round spermatids.

Disjunction of the Golgi apparatus and of the NDL was also observed in EM (Fig EV5A). Moreover, anti-GM130 staining in *Spink2$^{-/-}$* round spermatids appeared disseminated and punctuated, confirming the disorganization of the Golgi apparatus and indicating a fragmentation of the organelle (Fig 4D4–D6).

Interestingly, EM observations of *Spink2$^{-/-}$* round spermatids showed the presence of multivesicular bodies, a known biomarker of microautophagy (Li *et al*, 2012) (Fig EV5B). These latter structures strongly suggest that the absence of Spink2 activates an uncharacterized self-degradation pathway. Visual signs of the initial events of microautophagy occurring at the Golgi apparatus level are the engulfment of vesicles (Fig 4B2, black arrows) and the presence of already engulfed vesicles (Fig 4B2, black arrowhead). We note that the thorough examination of round spermatids on EM images did not reveal any detectable signs of morphological hallmarks of apoptosis such as chromatin condensation, fragmentation of the plasma membrane, and the presence of apoptotic bodies. Moreover, no differences in DNA damage were observed between WT and *Spink2$^{-/-}$* round spermatids when assessed by terminal deoxynucleotidyl transferase (TdT)-mediated deoxyuridine triphosphate (dUTP)-nick-end labeling (TUNEL) test (Appendix Fig S5). Altogether, these results suggest that the absence of Spink2 at the round-spermatid stage does not activate the apoptotic pathway.

## Rescue of acrosin-induced cell proliferation defects by coexpression with SPINK2

During spermatid differentiation, several enzymes, involved in sperm penetration through the protective layers surrounding the oocytes, accumulate in the acrosomal vesicle. Among these different enzymes, several proteases have been described to play a key role, including acrosin, believed to be the main acrosomal protease (Liu & Baker, 1993). Acrosin, a trypsin-like protease, is synthesized in the reticulum as a zymogen (proacrosin), transits through the Golgi apparatus, and accumulates in the acrosomal vesicle. Autoactivation of acrosin is pH-dependent and occurs at a pH > 6 (Meizel & Deamer, 1978) leading to sequential N-ter and C-ter cleavages of the proacrosin (46 kDa), eventually giving active forms of acrosin with lower weights of 20–34 kDa (Baba *et al*, 1989; Zahn *et al*, 2002). Since the pH of both the endoplasmic reticulum and the Golgi apparatus is greater than 6 (Rivinoja *et al*, 2012), we postulated that

Spink2, as a serine peptidase inhibitor, prevents acrosin autoactivation in these cellular compartments, thus preventing cellular stress induced by uncontrolled protease activation. Such stress would cause cellular defects including Golgi apparatus destabilization and defective acrosome biogenesis leading to spermatid differentiation arrest. To test this hypothesis, heterologous expressions of human C-terminus DDK-tagged proacrosin (ACR), SPINK2, or both were carried out in HEK293 cells and the kinetics of cell proliferation were followed using xCELLigence Real-Time Cell Analysis (RTCA) technology for the different conditions. It is worth noting that no members of the SPINK family are reported to be expressed in HEK293 cells. Analyses of kinetics showed that proacrosin expression quickly led to cell proliferation arrest and detachments in contrast to what was observed in the control condition (Fig 5A and B). Interestingly, cells showed a normal proliferation when SPINK2 was coexpressed with proacrosin (Fig 5A and B), therefore demonstrating that cell stress and damages induced by the proacrosin were prevented by SPINK2 coexpression. The presence of the different overexpressed proteins was verified in the different conditions by Western blotting using the SPINK2 antibody (Fig 5C), an anti-acrosin (Fig 5D), and the anti-DDK (Fig 5E) antibodies. In extracts of HEK293 cells transfected with proacrosin only and revealed with an anti-acrosin antibody (Fig 5D), two bands were present at around 46 and 34 kDa. The latter (red arrowhead) likely corresponds to the active form of acrosin resulting from the cleavage of proacrosin upon autoactivation. This band was not present when acrosin was coexpressed with SPINK2 or in non-transfected cells (control). Moreover, a closer inspection of the band around 46 kDa in the extracts of cells transfected with proacrosin only or proacrosin + SPINK2 shows that this band is of lower MW and was less intense in "acrosin" extract compared to "acrosin + SPINK2" cell extract, showing the process of successive cleavages occurring during proacrosin autoactivation (Zahn *et al*, 2002). Similar results were obtained with the anti-DDK antibody (Fig 5E). It is worth noting that anti-DDK antibody immunodecorates the zymogen form only and not the active form of acrosin because the C-terminus containing the DDK-tag is cleaved upon autoactivation. Western blot results thus demonstrate that coexpression of proacrosin with SPINK2 prevented its autoactivation. We can thus conclude that in the absence of a serine peptidase inhibitor, proacrosin can autoactivate and induces a cellular stress leading to

                                                       

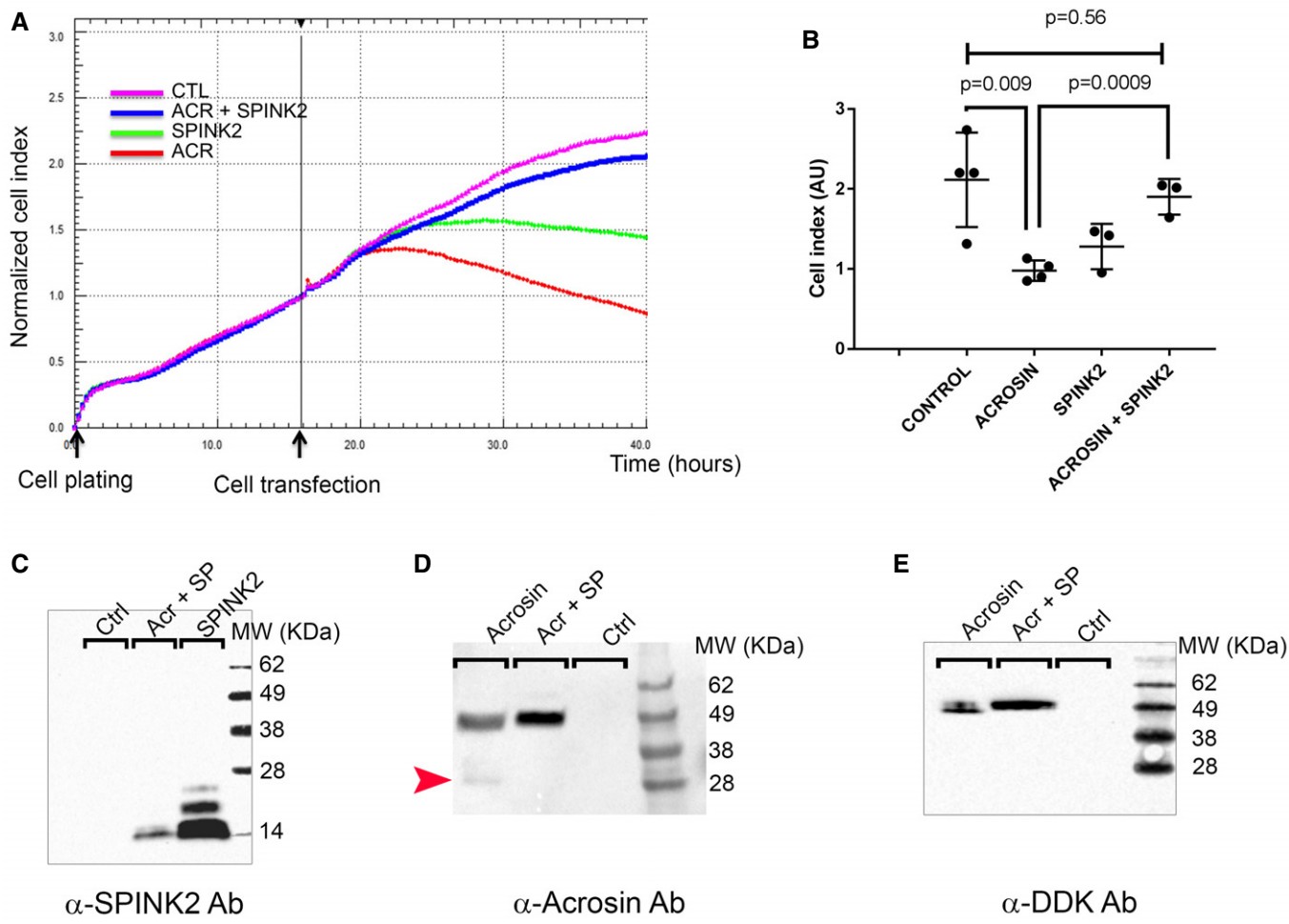

**Figure 5.   Heterologous expression of proacrosin in HEK293 cells induces acrosin activation and cell proliferation arrest, a phenotype rescued by SPINK2 coexpression.**

A   Representative kinetics of HEK293 cell proliferation measured with Real-Time Cell Analysis (RTCA) technology in different conditions as indicated. Each point corresponds to the mean of four technical replicates measured simultaneously. Black arrows indicate the time of cell plating ($t$ = 0 h) and introduction of the different plasmids in the cell chambers ($t$ = 16 h).

B   Scatter plots showing the mean and SD of the cell index measured at 40 h after plating (corresponding to cell proliferation and detachment) in different transfection conditions and measured for three independent biological replicates. Statistical differences were assessed using *t*-test, *P*-values as indicated.

C   Western blot using an anti-SPINK2 antibody showing the expression of SPINK2 in cell extracts of HEK293 cells transfected with different plasmids containing SPINK2 (SP) or acrosin and SPINK2.

D   Representative Western blot using an anti-acrosin antibody. In extracts of HEK293 cells transfected with proacrosin only (lane "acrosin"), two bands were observed, one at around 34 kDa and corresponding to the active form of acrosin (red arrowhead) and one at 46 kDa and corresponding to the zymogen form, whereas in extracts of HEK293 cells transfected with proacrosin and SPINK2 (lane "Acr + SP"), only the zymogen form was observed. Equal protein loading was verified by stain-free gel technology (Taylor & Posch, 2014) and Western blots against tubulin (Appendix Fig S6). Note that the zymogen form in lane "acrosin" has a slightly lower mass and that the band is less intense than that in lane "Acr + SP".

E   Representative Western blot using an anti-DDK antibody showing the expression of the proacrosin zymogen form in HEK293 cells transfected with different plasmids as indicated. Note that once more, the zymogen form in the lane "acrosin" has a slightly lower mass and that the band is less intense than that in lane "Acr + SP". Similarly, equal protein loading was verified by stain-free gel technology (Appendix Fig S6).

cell proliferation arrest and cell detachment, a phenotype similar to that observed in round spermatids from *Spink2*$^{-/-}$ males.

### SPINK2 haploinsufficiency induces sperm defects with incomplete penetrance in man

Only one additional subject, P105, was identified with a *SPINK2* heterozygous deleterious variant, and we cannot be sure that this variant is the cause of the patient's oligozoospermia. Two arguments could in fact suggest that *SPINK2* haploinsufficiency is not deleterious: (i) Br1 and Br2's father is SPINK2 heterozygous and has conceived six children spontaneously, and unfortunately, we could not obtain sperm samples to characterize this man's sperm parameters; and (ii) because heterozygous *Spink2*$^{+/-}$ male mice are fertile, they did not produce litters of reduced size (Fig 3A). We however carried out a detailed characterization of *Spink2*$^{+/-}$ and *Spink2*$^{+/+}$

sperm parameters to address the question of the impact of SPINK2 haploinsufficiency on mouse spermatogenesis. Heterozygous males displayed a significant increase in teratozoospermia (Fig 6A). Abnormal spermatozoa showed non-hooked heads, isolated heads, or a malformed base of the head (Fig 6B). Moreover, sperm motility of heterozygous males was impaired with lower total and progressive motility (Fig 6C). We note that the observed defects are very similar to those observed in the heterozygous patient P105 (Table EV1). We can therefore conclude that in mice, *SPINK2* haploinsufficiency induces asthenoteratospermia with no alteration of reproductive fitness, whereas in man it leads to oligoteratozoospermia with variable expressivity and infertility with an incomplete penetrance.

## Discussion

### SPINK family emerges as an important family for human genetic diseases

SPINK proteins are serine protease inhibitors containing one or several Kazal domains which interact directly with the catalytic domains of proteases blocking their enzymatic activity (Rawlings *et al*, 2004). The Kazal domain structure contains three disulfide bonds which are highly conserved. Different SPINK proteins are specifically expressed in different tissues and inhibit a number of serine proteases, such as secreted trypsin in the pancreas, acrosin in sperm, or kallikrein in the skin. Downregulation of the activity of different SPINK proteins leads to severe pathologies such as chronic pancreatitis and Netherton syndrome. In the pancreas, trypsin is produced as an inactive zymogen to prevent cell damage, yet the trypsinogen is occasionally able to autoactivate. This protease activity is then blocked by SPINK1. Chronic pancreatitis can be triggered by mutations of *SPINK1* that decrease or suppress its trypsin inhibitor function, leading to cell distress (Chen *et al*, 2000; Witt *et al*, 2000). In the skin, kallikrein-related peptidases are controlled by SPINK5 and unopposed kallikrein-peptidase activity due to *SPINK5* deficiency leads to Netherton syndrome, a severe skin disease (Furio & Hovnanian, 2014). SPINK6 and SPINK9 are also expressed in the skin, and altered expression levels are associated with atopic dermatitis or psoriasis (Redelfs *et al*, 2016). The other members of the SPINK family, including SPINK2, have not yet been associated with a human pathology. Here, we have clearly demonstrated that the absence of SPINK2 induces azoospermia, a severe infertility phenotype, emphasizing the importance of this family in human pathologies.

### Role of SPINK2 during spermiogenesis

We have shown that SPINK2 is located in the acrosomal vesicle in round spermatids and remains present in mature spermatozoa, suggesting that this protein is necessary for spermiogenesis and sperm survival. SPINK proteins are known to control protease activities in different tissues (Witt *et al*, 2000; Rawlings *et al*, 2004; Ohmuraya *et al*, 2012; Furio & Hovnanian, 2014) and since SPINK2 is located in the acrosome, it very likely neutralize acrosomal proteases before their release prior fertilization. Several proteases have been described to be present in the acrosome (Arboleda & Gerton, 1987; Kohno *et al*, 1998; Cesari *et al*, 2004). Among these,

acrosin (Acr) was the first to be described and is the acrosomal protein which has been the most studied. Acrosin is present in the acrosome as a zymogen called proacrosin (Huang-Yang & Meizel, 1975) which is predicted to be activated during the acrosome reaction (Brown & Harrison, 1978) upon a rise in acrosomal pH to 7 which induces pH-dependent proacrosin autoactivation (Baba *et al*, 1989). Before the acrosome reaction, at least two mechanisms prevent autoactivation: The first is the acrosomal acidic pH which is below 5, which blocks autoactivation of proacrosin (Meizel & Deamer, 1978); and the second is the presence in the sperm of a non-fully characterized proacrosin conversion inhibitor of 12 kDa which has been purified from boar acrosome (Kennedy *et al*, 1982). The presented results strongly suggest that this protein is in fact SPINK2. Proacrosin is however produced in the endoplasmic reticulum and transits through the Golgi apparatus, two cellular compartments with a pH of approximately 7 and 6.5, respectively. In these compartments, autoactivation of proacrosin is thus possible and would result in the release of active acrosin within these apparatuses. We therefore believe that SPINK2, which transits through the same cellular compartments, quenches this premature protease activity and prevents the described cascade of events leading to azoospermia. This hypothesis is supported by heterologous expression experiments: We have indeed demonstrated that proacrosin expression in HEK293 cells induces (i) autoactivation of proacrosin and (ii) cellular proliferation arrest and cell detachment. Moreover, cellular toxicity of proacrosin expression is prevented by SPINK2 coexpression, showing the ability of SPINK2 to inhibit acrosin activity.

One of the most striking effects of SPINK2 deficiency is the fragmentation of the Golgi apparatus, a key organelle for protein processing and translocation, in particular for membrane proteins. The notable strong desquamation of the germinal epithelium may be due to severe changes in membrane protein composition resulting from a defective Golgi apparatus function.

### Impact of SPINK2 deficiency

We have shown that the absence of SPINK2 in round spermatids leads to several subcellular defects targeting the process of proacrosomal vesicle formation by the Golgi apparatus. The observed abnormalities include the disorganization and delocalization of the Golgi apparatus, the presence of vesicles of various sizes, and the absence of proacrosomal vesicle fusion. The absence of SPINK2 likely allows proacrosin autoactivation within the reticulum and the Golgi apparatus compartments, leading to the above-described subcellular defects. It was previously shown that transgenic expression of porcine proacrosin in mice led to post-meiotic cell death and oligozoospermia, supporting the hypothesis that unbalanced expression of acrosin/Spink2 is deleterious (O'Brien *et al*, 1996). Interestingly, we have demonstrated that the cell responds to this stress by activating a microautophagy-like pathway: First, we showed that larger vacuoles engulfed small vacuoles, likely leading to the observed heterogeneity in vacuole size in the vicinity of the Golgi apparatus; and secondly, multivesicular bodies, a hallmark of microautophagy (Li *et al*, 2012), were clearly observed within Spink2$^{-/-}$ round spermatids, whereas they were never observed in WT. Furthermore, the lack of various SPINK proteins induces autophagy-induced cell death in regenerating Hydra (Chera *et al*, 2009)

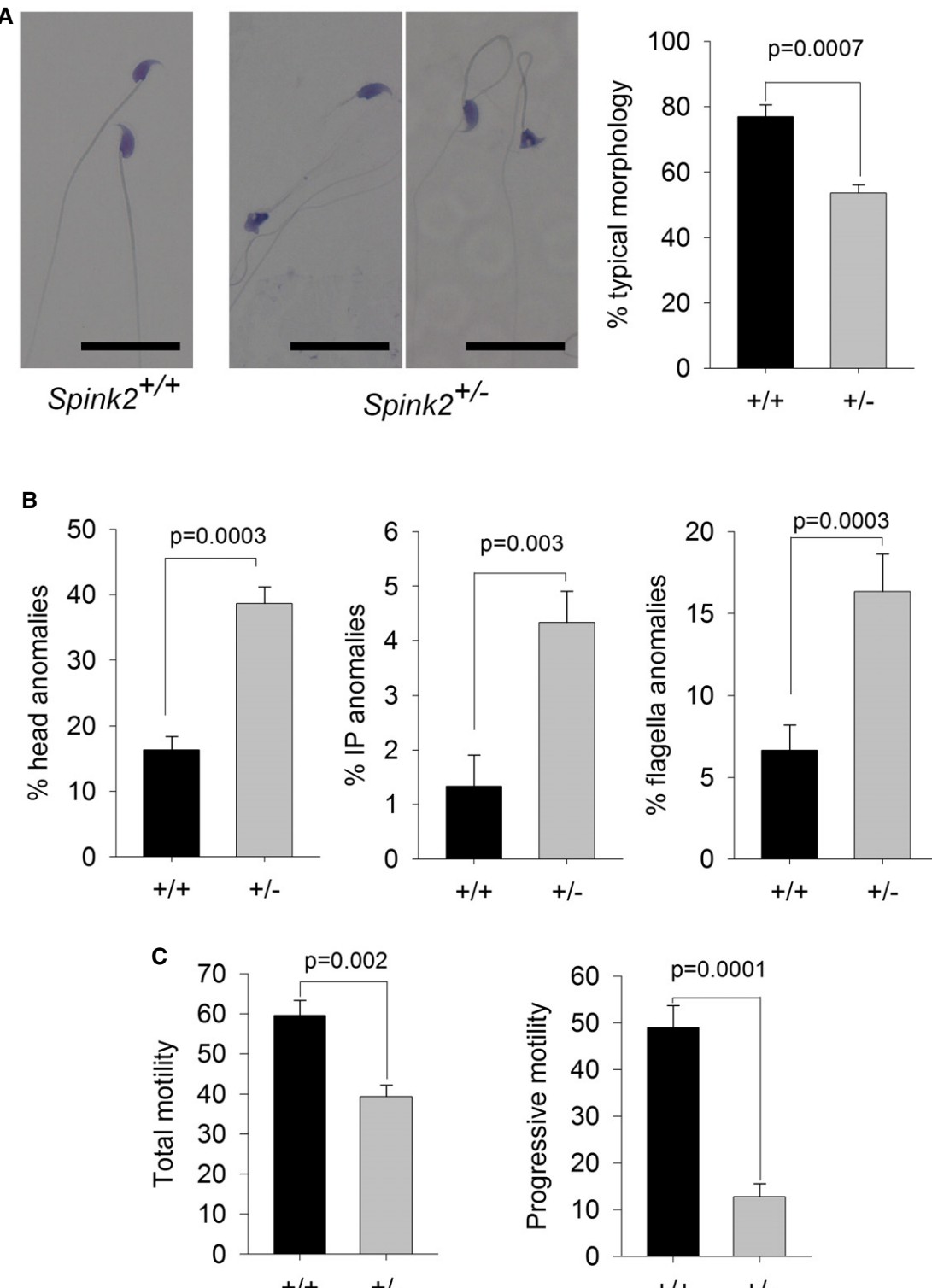

**Figure 6.  Sperm from *Spink2*$^{+/-}$ heterozygous mice exhibit morphological defects and low motility.**

A   Light microscopy analysis of sperm from *Spink2*$^{+/-}$ heterozygous mice reveals the presence of numerous non-typical forms of sperm. Scale bars, 25 μm. Graph on the right shows the mean ± SD percentage of defective sperm in WT (*n* = 3) and *Spink2*$^{+/-}$ mice (*n* = 3).

B   Anomalies were observed in the head and the mid- and principle pieces in WT and *Spink2*$^{+/-}$ mice (*n* = 3).

C   Total and progressive sperm motility were strongly decreased in *Spink2*$^{+/-}$ heterozygous mice (*n* = 5) in comparison with WT sperm (*n* = 5).

Data information: *n* represents the number of biological replicates, and for each replicate, more than 100 sperm were assessed per condition. Data are presented as mean ± SD. Statistical differences were assessed using *t*-test, *P*-values as indicated.

and was also described in newborn mice when *Spink3* (orthologue of *Spink1*) is mutated (Ohmuraya *et al*, 2005). Based on these results, it has been postulated that SPINK1/Spink3 could have the dual function of protease inhibitor and negative regulator of autophagy (Ohmuraya *et al*, 2012). Our results show that the absence of SPINK2 induces a microautophagy-like pathway in germ cells thereby further supporting this hypothesis.

### Oligozoospermia and azoospermia is a continuum correlated with *SPINK2* haploinsufficiency

We observed that in man, the presence of a homozygous *SPINK2* mutation leads to azoospermia while a heterozygous mutation can induce oligozoospermia suggesting that *SPINK2* haploinsufficiency can result in oligozoospermia. In mice, we showed that the complete absence of the protein leads to azoospermia. We also showed that heterozygous animals have terato-astheno-zoospermia but with no obvious decrease in sperm number and no impact on fertility. A previous study carried out in a different mouse hypomorphic mutant line showed that a significant inactivation of *Spink2* (likely in excess of 90%) led to a reduction by half of sperm number within the epididymis and a five-fold increase in morphologically abnormal spermatozoa. Male mice also exhibited a reduced fertility and produced litters of reduced size with an average of 5.19 pups by litter compared to 8.56 in controls (Lee *et al*, 2011). These results and the results presented therein thus confirm that in mice, the severity of the phenotype is dependent on *Spink2* expression levels and that there is a phenotypic continuum ranging from (i) azoospermia in the complete absence of the protein (ii) to teratozoospermia and oligozoospermia associated with subfertility when only a fraction of the protein is present and finally (iii) to astheno-teratozoospermia with no impact on fertility when half of the protein is present. These observations in mice strongly support the notion that SPINK2 heterozygous mutations in man will impact spermatogenesis with a variable effect on fertility. We identified only one heterozygous mutation out of 611 analyzed patients indicating that *SPINK2* variants are very rare, likely because heterozygous variants underwent a strong negative selection during evolution. This hypothesis is supported by data from the ExAC database which indicate that the *SPINK2* gene has a high probability of loss of function intolerance (pLI = 0.72).

The testis is the organ which expresses the highest number of tissue-specific transcripts ($n > 500$) (Feig *et al*, 2007; Dezso *et al*, 2008) and altered spermatogenesis has been observed in knockout mouse models for more than 388 genes (Massart *et al*, 2012). It is therefore expected that NOA is genetically highly heterogeneous and that few patients carry causal defects on the same gene. Due to the involvement of the corresponding proteins in multiple phases of spermatogenesis, the causes of azoospermia are numerous and involve genes controlling spermatogonial self-renewal, meiosis, and spermiogenesis. Here, we have confirmed that alterations of spermiogenesis do not only lead to teratozoospermia as described several times previously (Dieterich *et al*, 2007; Harbuz *et al*, 2011; Ben Khelifa *et al*, 2014) but also to azoospermia. The vast majority of patients with an altered spermatogenesis can be treated with IVF or by the direct injection of a sperm into the oocyte (ICSI). Most patients with NOA however cannot benefit from ICSI-IVF treatments. Identifying the genetic defects responsible for NOA and characterizing their molecular pathogeny will provide a basis for the

development of therapeutic solutions tailored to the patient. In this particular case, we have shown that SPINK2 deficiency can induce azoospermia and demonstrated that unrestricted acrosomal protease activity induces the arrest of spermiogenesis. Moreover, we provided evidence that this process activates a microautophagy-like pathway. As we have shown that the pool of undifferentiated spermatogonia is not affected, we can envisage a method of treatment targeting protease activity using a protease inhibitor, as is done for chronic pancreatitis caused by SPINK1 deficiency (Kambhampati *et al*, 2014).

# Materials and Methods

### Patients and biological samples

Human sperm were obtained from patients consulting for diagnosis or assisted reproductive techniques at the fertility center of the Grenoble University Hospital. All patients signed an informed consent for use of part of their samples in research programs respecting the WMA declaration of Helsinki. The samples were then stored in the CRB Germetheque (certification under ISO-9001 and NF-S 96-900) following a standardized procedure. Consent for CRB storage was approved by the CPP Sud-Ouest of Toulouse (coordination of the multisite CRB Germetheque). The storage and transfer authorization number for the CRB Germetheque is AC2009-886. The scientific and ethical board of the CRB Germetheque approved the transfer of the semen samples for this study. Additional DNA samples from patients with azoospermia and oligozoospermia were obtained from the CHU of Grenoble, Saint Etienne, and Marseille. All patients gave their informed consent for the anonymous use of their leftover samples. Brothers Br1 and Br2 are French citizens from a traveling group originating from Romania but whose recent ancestors lived in Spain and the south of France. Subject P105 is also a French citizen with eastern ascendants (from Russia).

### Exome sequencing and bioinformatic analysis

Genomic DNA was isolated from saliva using Oragene saliva DNA collection kit (DNA Genotek Inc., Ottawa, Canada). Exome capture was performed using NimbleGen SeqCap EZ Kit version 2 (Roche). Sequencing was conducted on an Illumina HiSeq 2000 instrument with paired-end 76-nt reads. Sequence reads were aligned to the reference genome (hg19) using MAGIC (SEQC/MAQC-III Consortium, 2014). Duplicate reads and reads that mapped to multiple locations in the exome were excluded from further analysis. Positions with sequence coverage below 10 on either the forward or reverse strand were excluded. Single nucleotide variations (SNV) and small insertions/deletions (indels) were identified and quality-filtered using in-house scripts. The most promising candidate variants were identified using an in-house bioinformatics pipeline. Variants with a minor allele frequency > 5% in the NHLBI ESP6500 or in 1000 Genomes Project phase 1 datasets, or > 1% in ExAC, were discarded. We also compared these variants to an in-house database of 56 control exomes. All variants present in a homozygous state in this database were excluded. We used Variant Effect Predictor (VEP) to predict the impact of the selected variants. We only retained variants impacting splice donor/acceptor sites or

    

causing frameshift, in-frame insertions/deletions, stop gain, stop loss, or missense variants except those scored as "tolerated" by SIFT (sift.jcvi.org) and as "benign" by PolyPhen-2 (genetics.bwh.harvard.edu/pph2). All steps from sequence mapping to variant selection were performed using the ExSQLibur pipeline (https://github.com/tkaraouzene/ExSQLibur). Our datasets were obtained from subjects who have consented to the use of their individual genetic data for biomedical research, but not for unlimited public data release. Therefore, we submitted it to the European Genome-phenome Archive, through which researchers can apply for access of the raw data under the accession number EGAD00001003326.

### Sanger sequencing

Sanger sequencing of the four *SPINK2* exons and intron borders was carried out using the primers described in Appendix Table S3. Thirty-five cycles of PCR amplification were carried out with a hybridization temperature of 60°C. Sequencing reactions were performed using BigDye Terminator v3.1 (Applied Biosystems). Sequence analyses were carried out on ABI 3130XL (Applied Biosystems). Sequences were analyzed using seqscape software (Applied Biosystems).

### RT–PCR and quantitative real-time PCR

Total RNA from various tissues including testes from three WT and homozygous KO mice was extracted using the mirVana™ PARIS™ Kit (Life Technologies®) according to the manufacturer's protocol. Human cDNAs were obtained from Life Technologies® mRNA.

Reverse transcription was carried out with 5 µl of extracted RNA (~500 ng). Hybridization of the oligo dT was performed by incubating for 5 min at 65°C and quenching on ice with the following mix: 5 µl RNA, 3 µl of poly-T-oligo primers (dT) 12–18 (10 mM; Pharmacia), 3 µl of the four dNTPs (0.5 mM, Roche Diagnostics) and 2.2 µl of $H_2O$. Reverse transcription then was carried out for 30 min at 55°C after the addition of 4 µl of 5× buffer, 0.5 µl RNase inhibitor, and 0.5 µl Transcriptor reverse transcriptase (Roche Diagnostics). One microliter of the obtained cDNA mix was used for the subsequent PCR. Primers are described in Appendix Table S4.

A specific region of the transcript was amplified using a StepOnePlus™ Real-Time PCR System (Life Technologies®) with Power SYBR® Green PCR Master Mix (Life Technologies®) according to the manufacturer's protocol. PCR without template was used as a negative control to verify experimental results. The sequence for oligonucleotide primers used and their product sizes are summarized in Appendix Table S5.

After amplification, the specificity of the PCR was determined by both melt-curve analysis and gel electrophoresis to verify that only a single product of the correct size was present. Quantification of the fold change in gene expression was determined by the relative quantification method ($2^{-\Delta\Delta C_T}$) using the beta-actin gene as a reference. Data are shown as the average fold increase ± standard error of the mean.

### Primary antibodies

SPINK2 rabbit polyclonal antibody was from Sigma-Aldrich (HPA026813) and used at 1/1,000 for Western blot analysis. Sperm protein Sp56 and Golgi matrix protein GM130 (610822) mouse monoclonal antibodies were from QED Bioscience Inc. (used at 1/800 and 1/200, respectively). Promyelocytic leukemia zinc finger protein PLZF rabbit polyclonal antibody (Sc-22839) was from Santa Cruz Biotechnology. Dpy19l2 antibodies were produced in rabbit as polyclonal antibodies raised against RSKLREGSSDRPQSSC and CTGQARRRWSAATMEP peptides corresponding to amino acids 6–21 and 21–36 of the N-terminus of mouse Dpy19l2 (Pierre *et al*, 2012). DDK antibody was from OriGene (TA50011) or Sigma-Aldrich (FLAG® M2F1804) and used at 1/10,000 for Western blot analysis. Acrosin antibody was previously described (Gallo *et al*, 1991) and is a gift from Denise Escalier.

### Western blot analysis

HEK293 cells were lysed in 25 mM Tris pH 7.4, 5 mM EDTA, 1% Triton X-100, and complete protease inhibitor cocktail (Roche) and were then centrifuged. After centrifugation at 20,000 *g* for 15 min at 4°C, the soluble supernatant was conserved and subjected to SDS–PAGE. The protein concentration from supernatants was quantified by the bicinchoninic acid assay (BCA assay) using bovine serum albumin as a standard. Sample concentrations were adjusted and mixed with 1× high-SDS sample buffer (4% SDS, 62 mM Tris–HCl pH 6.8, 0.1% bromophenol blue, 15% glycerol, 5% ß-mercaptoethanol) and separated using 4–20% SDS mini-PROTEAN® TGX Stain-Free™ Precast Gels (Bio-Rad) or 10 and 20% polyacrylamide–SDS gels and transferred into PVDF membranes (Millipore, 0.2 µm) using Trans-Blot® Turbo™ Blotting System and Midi Transfer Packs (Bio-Rad). The membranes were blocked in 5% non-fat dry milk in PBS/0.1% Tween and incubated for 1 h at room temperature with the primary antibody, followed by 45-min incubation with a species-matched horseradish peroxidase-labeled secondary antibody (1/10,000) (Jackson ImmunoResearch). Immunoreactivity was detected using chemiluminescence detection kit reagents (Luminata; Millipore) and a ChemiDoc Station (Bio-Rad).

### Real-time cell analysis

The growth, proliferation, and adhesion kinetics of HEK293 cells were determined using RTCA technology (ACEA Biosciences, San Diego, CA, USA). Fifty microliters of DMEM supplemented with 10% HI-FBS and 50 µg/ml gentamicin (cell culture medium) was loaded in each well of the E-plate 96 (gold-microelectrode array integrated E-plate; ACEA Biosciences). E-plate 96 was then connected to the system to obtain background impedance readings. Around $1.5 \times 10^4$ cells in 50 µl were added to the wells containing 50 µl of culture medium. The E-plates were placed on the RTCA SP Station located in a 37°C, 5% $CO_2$ tissue culture incubator for continuous impedance recording. The cell index values measured by continuous impedance recordings every 5 min are proportional to the number of adherent cells. After 16–17 h, cells were transfected as described below, and for each of the conditions, four replicates were done. The assay was conducted for 40 h.

### Mice

All animal procedures were run according to the French guidelines on the use of animals in scientific investigations with the approval of

the local ethical committee (Grenoble-Institut des Neurosciences—ethical committee, study agreement number 004). Mice were euthanized by cervical dislocation.

The Spink2tm1.1 (KOMP)Vlcg mouse strain used for this research project was created from ES cell clone Spink2_AG5_M7, generated by Regeneron Pharmaceuticals, Inc. and made into live mice by the KOMP Repository (www.komp.org) and the Mouse Biology Program (www.mousebiology.org) at the University of California Davis. The methods used to create the VelociGene targeted alleles have been published (Valenzuela *et al*, 2003). They were then reared by the Mouse Clinical Institute—MCI—located in Strasbourg as part of the "knockout mouse project". The colony used in this study was initiated from two couples consisting of heterozygous females and males. Mice were housed with unlimited access to food and water and were sacrificed after 8 weeks of age (the age of sexual maturity).

## Genotyping

DNA for genotyping was isolated from tail biopsies. Tail biopsies (2 mm in length) were digested in 200 μl of DirectPCR Lysis Reagent (Tail) (Viagen Biotech Inc, CA, USA) and 0.2 mg of proteinase K for 12–15 h at 55°C followed by 1 h at 85°C for proteinase K inactivation. The DNA was directly used for PCRs. Multiplex PCR was done for 35 cycles, with an annealing temperature of 58°C, and an elongation time of 60 s at 72°C. PCR products were separated by 2% agarose gel electrophoresis. Genotypes were determined according to the migration pattern. Primers are described in Appendix Table S6.

## Phenotypic analysis of mutant mice

To test fertility, pubescent $Spink2^{-/-}$ males (8-week-old) were mated with WT females.

To determine sperm concentration, sperm samples were collected from the cauda epididymis and vas deferens of 8-week-old males, and sperm number was determined using a hemocytometer under a light microscope.

## Sperm motility analysis

Experiments were performed on a CASA CEROS v.12 (Hamilton Thorne Biosciences, Beverly, MA, USA) using Leja double-chamber slides (Leja Products B.V., the Netherlands) for standard count with 100 μm depth. After epididymal extraction, sperm cells were allowed to swim for 10 min at 37°C and then were immediately analyzed. At least 150 cells were analyzed per sample with the following parameters: acquisition rate: 60 Hz; number of frames: 45; minimum contrast: 50; minimum cell size: 5; low static-size gate: 0.3; high static-size gate: 1.95; low static-intensity gate: 0.5; high static-intensity gate: 1.3; minimum elongation gate: 0; maximum elongation gate: 87; and magnification factor: 0.7. The motility parameters measured were curvilinear velocity (VCL), straight-line velocity (VSL), average path velocity (VAP), and amplitude of lateral head displacement (ALH). Motile sperm were defined by VAP > 1 and progressive sperm were defined by VAP > 30 and VSL/VAP > 0.7.

## Histological analysis

To analyze testicular integrity, testes from adult $Spink2^{+/+}$ and $Spink2^{-/-}$ mice were fixed by immersion in 4% paraformaldehyde (PFA) for 14 h, embedded in paraffin, and sectioned (4 μm). For histological analysis, after being deparaffinized slides were stained with hematoxylin and eosin or by the PAS technique. The colored sections were digitized at ×40 magnification through an Axioscope microscope (Zeiss, Germany) equipped with a motorized X–Y-sensitive stage. For sperm morphology analysis, sperm were washed twice in PBS and then displayed over slides, dried at room temperature, and then fixed in 75% ethanol for Harris–Shorr staining. At least 100 sperm cells were analyzed per sample.

## Testicular germ cell dissociation

C57BL/6 male or *Spink2* KO mice (8-week-old) were euthanized by cervical dislocation. The testes were surgically removed and placed in PBS (at room temperature). The tunica albuginea was removed from the testes with sterile forceps and discarded. Then, the testes were incubated in 1 mg/ml of collagenase solution in EKRB cell buffer containing (in mM) 2 $CaCl_2$, 12.1 glucose, 10 HEPES, 5 KCl, 1 $MgCl_2$, 6 Na-lactate, 150 NaCl, 1 $NaH_2PO_4$, and 12 $NaHCO_3$ pH 7, and agitated horizontally at a maximum of 120 rpm for 30 min at 25°C. The dispersed seminiferous tubules were then washed with PBS and cut thinly. Cells were dissociated by gentle pipetting filtered through a 100-μm filter and then pelleted by centrifugation at 500 *g* for 7 min. Cells were resuspended in 1 ml PBS, fixed with 4% PFA solution, washed with PBS, and finally layered onto polylysine-coated slides.

## Immunohistochemistry

Mice were anesthetized by intraperitoneal injection of a ketamine/xylazine cocktail (87.5 mg/kg ketamine and 12.5 mg/kg xylazine) and sacrificed through intracardiac perfusion of PFA (4%). The testes and epididymides were removed and fixed for a further 8 h before paraffin embedding and sectioning. Mature sperm cells were obtained for analysis through mechanical dilaceration of the epididymis. Sperm cells were fixed in 4% PFA for 1 min and washed in PBS before being spotted onto poly-L-lysine-pre-coated slides. Spermatogenic cells of the round-spermatid stage were purified by unit gravity sedimentation from a spermatogenic cell suspension obtained from sexually mature males as described in Yassine *et al* (2015).

For immunofluorescence experiments, heat-induced antigen retrieval was performed by boiling slides immersed in either 0.01 M sodium citrate buffer–0.05% Tween-20, pH 6.0, or 10 mM Tris base–1 mM EDTA solution–0.05% Tween-20, pH 9.0, for 15–25 min. Sections were blocked in 2% goat serum–0.1% Triton X-100 for 1 h at RT and incubated with primary antibodies overnight at 4°C. The slides were then washed and incubated with secondary antibody (DyLight 549-conjugated goat anti-mouse IgG or DyLight 488-conjugated goat anti-rabbit IgG, Jackson ImmunoResearch) and Hoechst 33342 for 2 h at RT, rinsed, and mounted with Dako mounting medium (Life Technology). Images were taken by confocal microscopy (Zeiss LSM 710) and processed using Zen 2009 software.

## Electron microscopy (EM)

Adult male mice were anesthetized and fixed by intracardiac injection with 2% glutaraldehyde and 2.5% PFA in 0.1 M cacodylate, pH 7.2. For morphological analysis, samples were fixed with 2.5% glutaraldehyde in 0.1 M cacodylate buffer pH 7.4 over 24 h at room temperature. Samples were then washed with buffer and post-fixed with 1% osmium tetroxide and 0.1 M cacodylate pH 7.2 for 1 h at 4°C. After extensive washing with water, cells were further stained with 1% uranyl acetate pH 4 in water for 1 h at 4°C before being dehydrated through graded alcohol (30%–60%–90%–100%–100%–100%) and infiltrate with a mix of 1/1 epon/alcohol 100% for 1 h and several baths of fresh epon (Flukka) during 3 h. Finally, samples were embedded in a capsule full of resin that was left to polymerize over 72 h at 60°C. Ultrathin sections of the samples were cut with an ultramicrotome (Leica), and the sections were post-stained with 5% uranyl acetate and 0.4% lead citrate before being observed with an electron microscope at 80 kV (JEOL 1200EX). Images were acquired with a digital camera (Veleta; SIS, Olympus), and morphometric analysis was performed with iTEM software (Olympus).

## Cell culture and transfection

Mycoplasma-free HEK293 cells were a gift from A. Andrieux from Grenoble Neuroscience Institute and grown in Dulbecco's modified Eagle's medium supplemented with 10% FBS (Invitrogen, France) and 50 μg/ml gentamicin (Sigma) in a 37°C, 5% $CO_2$ cell culture incubator and transiently transfected with Cter-DDK-tagged human acrosin (RC214256; OriGene, Rockville, MD, USA) and/or human *SPINK2* (RC205388; OriGene) and/or human c.1A>T mutated *SPINK2*-containing pCMV6 plasmids, using JetPRIME Transfection Reagent (Polyplus, France) according to the manufacturer's instructions. For immunochemistry experiments, transfected cells were fixed with 4% PFA 2 days after transfection.

## DNA strand breaks

Sections were permeabilized using a 0.1% (v/v) Triton X-100 and 0.1% (w/v) sodium citrate in 1× PBS for 2 min and labeled by terminal deoxynucleotidyl transferase-mediated deoxy-UTP nick-end labeling (TUNEL) according to the Roche protocol of the *In Situ* Cell Detection Kit (Roche Diagnostics, Mannheim, Germany). Nuclei were counterstained in a 0.5 μg/ml Hoechst solution for 3 min, washed in PBS for 3 min, and mounted with DAKO mounting medium.

## Statistical analyses

*n* represents the number of biological replicates. For sperm analyses, for each replicate, more than 100 sperm were assessed per condition. Statistical analyses were performed with SigmaPlot 10 and GraphPad Prism 7. *t*-Tests were used to compare WT and KO samples. Data represent mean ± SEM or SD, as indicated. Statistical tests with a two-tailed *P*-value ≤ 0.05 were considered significant.

**Expanded View** for this article is available online.

### The paper explained

#### Problem

Infertility concerns one in seven couples and is usually addressed by performing *in vitro* fertilization (IVF) often by injecting spermatozoa directly into the oocytes by intracytoplasmic sperm injection (ICSI). Some men have a non-obstructive azoospermia (NOA), caused by a deficient spermatogenesis, and have no spermatozoa in the ejaculate. In some cases, a testicular biopsy can be performed in hope of finding some mature spermatozoa that will be used for ICSI, but most men with NOA will not be able to have biological children. It is believed that most cases of NOA are caused by a genetic factor, but a diagnosis is obtained for only approximately 20% of patients.

#### Results

We performed exome sequencing on two brothers with NOA and identified a homozygous mutation in the *SPINK2* gene coding for a serine protease inhibitor believed to target the acrosin, the main protease of the acrosome, a large vesicle located to the anterior part of the spermatozoa and containing an enzyme mix necessary to perforate the zona pellucida of the oocyte to achieve fertilization. Mouse study allowed to observe that homozygous KO male also had NOA, confirming the human diagnostic. Germ cells could go through meiosis but were blocked at the round-spermatid stage. We further observed that in the round spermatids, in the absence of SPINK2, the acrosin could autoactivate during its transit through the endoplasmic reticulum and the Golgi apparatus leading to a disorganization of the Golgi and its inability to form the acrosome and a block at the round-spermatid stage. We further demonstrate that the presence of a heterozygous SPINK2 mutation was also deleterious leading to the production of sperm with variable levels of anomalies.

#### Impact

We identified a new gene leading to male infertility permitting to improve the diagnostic efficiency for NOA patients. We demonstrate that whole-exome sequencing is an efficient technique to identify new infertility genes and to realize a diagnostic for affected men. We showed that the control of proteases by antiproteases, and in particular by SPINK2, is critical during spermiogenesis and demonstrate that the SPINK gene family is involved not only in pancreatitis or skin disease but also in male infertility.

## Acknowledgements

We thank the GIN electron microscopy platform and Anne Bertrand, and the IAB microscopy platform and Alexei Grichine and Jacques Mazzega for their technical help. We thank Myriam Dridi for her work on HEK cells and antibody validation, Jean Pascal Hograindleur for his help for CASA experiments and Denise Escalier for her generous gift of human anti-acrosin antibody. This work was mainly supported by the French research agency (ANR) within the 2009 Genopat program for the ICG2I project "Identification and characterization of genetic causes of male infertility" to PR and CA. Support was also obtained from the Fondation Maladies Rares (FMR) for the project R16070CC, "Identification of genetic causes of human NOA".

## Author contributions

PFR and CA designed the study, supervised all laboratory work, and wrote the manuscript. They have full access to all of the data in the study and take responsibility for the integrity of the data and its accuracy. All authors read, corrected, and made a significant contribution to the manuscript. Z-EK, TK, AA-Y, CB, MG, NT-M, and CC produced and analyzed the genetic data, and Z-EK, MC-K, AA-Y, and ASV performed immunohistochemistry (IF) experiments. SPB, JE and EL performed Western blot experiments and real-time cell

analyses. Z-EK and GM performed sperm analyses; KP-G and ASV performed the electron microscopy; and BD, IA-S, MM, CM-G, SN, VS, MB, FB, JF, and SH provided clinical samples and data and supplied biological materials.

## Conflict of interest

The authors declare that they have no conflict of interest.

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
