## [Review Process File · EMBO Molecular Medicine]

SPINK2 deficiency causes infertility by inducing sperm defects in heterozygotes to azoospermia in homozygotes

Zine-Eddine Kherraf, Marie Christou-Kent, Thomas KaraouzÈne, Amir Amiri-Yekta, Guillaume Martinez, Alexandra S Vargas, Emeiline Labert, Christelle Borel, BÉatrice Dorphin, Isabelle Aknin-Seifer, Michael Mitchell, Catherine Metzler-Guillemain, Jessica Escoffier, Serge Nef, Marianne Grepillat, Nicolas Thierry-Mieg, Veronique Satre, Marc Bailly, Florence Boïterelle, Karin Pernet-Gallay, Sylviane Hennebicq, Julien Faure, Serge P Bottari, Charles Coutton, Pierre F Ray, Christophe Arnoult

Corresponding author: Pierre Ray, Chu de Grenoble & Christophe Arnoult, Université Grenoble Alpes

Review timeline:

Submission date:	13 December 2016
Editorial Decision:	17 January 2017
Revision received:	06 April 2017
Editorial Decision:	10 April 2017
Revision received:	14 April 2017
Accepted:	26 April 2017

Transaction Report:

Editor: Céline Carret

1st Editorial Decision

17 January 2017

Thank you for the submission of your manuscript to EMBO Molecular Medicine. We have now heard back from the two referees whom we asked to evaluate your manuscript.

You will see that the referees find your study well conducted and interesting. While referee 2 is supportive of publication in its current form, referee 1 suggests adding some mechanism, increasing clinical relevance by making the same point mutation in the mouse as in patients, and eventually back-crossing the new mouse to acrosin-KO animals. In principle, we agree that all these experiments would greatly improve the paper, however we also believe that this might be too much to ask for a revision. From our point of view, we would like to encourage you to address the CRISPR/Cas9 experiment to increase clinical relevance or to add molecular mechanism to better understand the data.

Given the balance of these evaluations, we feel that we can consider a revision of your manuscript if you can address the issues that have been raised within the time constraints outlined below. Please note that it is EMBO Molecular Medicine policy to allow only a single round of revision and that, as acceptance or rejection of the manuscript will depend on another round of review, your responses should be as complete as possible. EMBO Molecular Medicine has a "scooping protection" policy, whereby similar findings that are published by others during review or revision are not a criterion

for rejection. Should you decide to submit a revised version, I do ask that you get in touch after three months if you have not completed it, to update us on the status.

Revised manuscripts should be submitted within three months of a request for revision; they will otherwise be treated as new submissions, except under exceptional circumstances in which a short extension is obtained from the editor.

Please read below and consult our guidelines for important editorial formatting. I would also encourage you to limit the number of main figures by pooling figures that are not already too crowded. Figures should be provided in portrait format. Please remember that you can add up to 5 EV figures, the rest can go into Appendix supplementary figures (see guidelines).

I look forward to receiving your revised manuscript.

***** Reviewer's comments *****

Referee #1 (Remarks):

Kherraf et al., report non-obstructive azoospermia and male infertility in two brothers of a consanguineous family. Each brother has the same c.56-C>G mutation in the SPINK2 gene that encodes an 84 aa Kazal type 2 serine protease inhibitor. Two abnormal splice variants were predicted and confirmed by RT-PCR and DNA sequence in one of the brothers. By screening 611 non-consanguineous patients with a- or oligospermia, an additional SPINK2 variant (c.1A>T) which mutates the start codon was identified. The patient, heterozygous for the mutation, was oligospermic with defects in the acrosome and sperm head-neck morphology. One son was born to this patient who sought medical attention for subsequent infertility.

The association of defective SPINK2 and fertility was further explored in a mouse model system in which a gene trap (tm1.1) mutated the Spink2 gene. Male, but not female mice, were infertile with loss of mature sperm in their epididymides. Using immunofluorescence and electron microscopy, the authors defined a defect in acrosome biogenesis, Golgi apparatus fragmentation and early arrest of round spermatid differentiation.

Comments:

Overall, this is a careful study focused on documenting that the association of a male infertility with a genetic mutation in SPINK2 is causal using KO mice as a model system.

1. Lee et al., (JBC 286:29108, 2011) previously reported a gene-trap mutation of Spink2 lacking exon 1 with decreased levels of protein and oligospermia, but a rather modest decrease in litter size (5.2 vs. 8.6 pups/litter). The discussion of this earlier study should be expanded and integrated with the current observations.
2. Although the phenotype of Spink2^{-/-} mice support causality of human phenotype, the same point mutation observed in humans made by CRISPR/Cas9 in mice would make the case more compelling.
3. It would be of interest to cross the Spink2^{-/-} mice with Acrosin^{-/-} mice which should rescue the Spink2^{-/-} phenotype if SPINK2's sole physiologic function is to inhibit acrosin during spermatogenesis.
4. The descriptive morphology of abnormalities of the Golgi and acrosome do not provide molecular mechanistic insight as to why/how the absence of SPINK2, an inhibitor acrosin, results in the observed phenotype.
5. Fig. 1B should have age-matched fertile controls.
6. The hypothetical transcript in Fig. 3B should be confirmed by RT-PCR and sequence.

Referee #2 (Remarks):

By using exome sequencing in two azoospermic brothers, born from related parents, the authors found a homozygous mutation in SPINK2 gene encoding a serine protease inhibitor. Mutations of this gene seem to be rare since only one oligozoospermic patient carrying a heterozygous mutation was subsequently found among 611 infertile men.

Then, they obtained and studied Spink2 KO mice and observed that males were completely infertile whereas fertility in females was not impaired. Histological analysis of testis in Spink2 $-/-$ male mice revealed a post-meiotic blockage at the early round spermatid stage. Immunohistochemical studies, in human and mouse testis, showed that the protein SPINK2 was present within the acrosomal vesicle in round spermatids and then in the acrosome in mature spermatozoa. In Spink2 $-/-$ male mice, electron microscopy revealed that proacrosomal vesicle fusion was impaired and that Golgi apparatus was fragmented. The authors showed also that the Golgi was localized randomly around the spermatid nucleus instead of being face to a particular area of the nuclear membrane where acrosome is supposed to be linked to the nucleus. The observation of multivesicular bodies within the cytoplasm of Spink2 $-/-$ spermatids led the authors to suggest that the lack of SPINK2 protein could activate a self-degradation process, named microautophagy, independent from apoptosis. Although Spink2 $+/-$ male mice are fertile, the authors analyzed in details the sperm parameters of these mice and found that the rate of teratozoospermia was significantly increased whereas mobility was decreased. These abnormal parameters were similar to those observed in the oligozoospermic patient carrying a heterozygous mutation which led the authors to conclude that haploinsufficiency of SPINK2 can lead to oligoteratozoospermia in man.

The discussion deals with the role of SPINK family proteins in protecting secretory cells against damages due to enzymes they produce. Particularly, the role of SPINK2 in protecting spermatids against acrosomal enzymes is treated and its relationships with the phenotype of the patients who were studied is convincing.

In conclusion, this is an interesting paper dealing with a rare event, the discovery of a new gene involved in human spermatogenesis. There is no major modification to make and it can be published in state.

Minor comments:

- At the end of results, it is written that "SPINK2 haploinsufficiency is not be deleterious". Please, correct.

- At the beginning of the discussion, the sentence "chronic pancreatitis or Netherton syndrome" suggests that this syndrome corresponds to the pancreatitis when it's a dermatologic disease, as defined a few lines later. Please modify by "chronic pancreatitis and Netherton syndrome"

- In the next paragraph, acrosomal activity is released just before fertilization, not at fertilization. Two lines later, a space is missing before "among".

1st Revision - authors' response

06 April 2017

Point by point answer to the reviewers:

Referee #1 (Remarks):

Comments:

Overall, this is a careful study focused on documenting that the association of a male infertility with a genetic mutation in SPINK2 is causal using KO mice as a model system.

- Lee et al., (JBC 286:29108, 2011) previously reported a gene-trap mutation of Spink2 lacking exon 1 with decreased levels of protein and oligospermia, but a rather modest decrease in litter size (5.2 vs. 8.6 pups/litter). The discussion of this earlier study should be expanded and integrated with the current observations.*

This paper was already cited page 12. We have now expanded the description of this paper in the discussion section.

2. *Although the phenotype of Spink2^{-/-} mice support causality of human phenotype, the same point mutation observed in humans made by CRISPR/Cas9 in mice would make the case more compelling.*

In our manuscript we report two variants: c.56-3C>G which was homozygous in the two brothers and c.1A>T which was heterozygous in a patient with oligoteratospermia. We agree that the replication of the exact mutations in mice would be ideal but these are expensive and time-consuming experiments, which are not compatible with the timeframe of this revision.

Regarding the first mutation we demonstrated that it induces two abnormal splices, which are not expected to produce any functional proteins. The effect of this mutation is therefore perfectly comparable to the null mutation created in the KO animal.

Regarding the second mutation we have added some additional results. We introduced this mutation (c.1A>T) in a plasmid and tested the production of a putative mutant protein in a heterologous expression system (HEK293 cells). Using two antibodies, no signal corresponding to a mutant SPINK2 protein was observed. This experiment is now shown in Fig 3C and demonstrates that this mutation also corresponds to a null mutation and thus that patient 105 is SPINK2 haploinsufficient.

3. *It would be of interest to cross the Spink2^{-/-} mice with Acrosin^{-/-} mice which should rescue the Spink2^{-/-} phenotype if SPINK2's sole physiologic function is to inhibit acrosin during spermatogenesis.*

Although this experiment would be very informative, such an experiment is not compatible within the time windows of this revision and was thus not possible.

To address this issue we however carried out some extensive cell work. We demonstrate that Acrosin (ACR) overexpression in HEK cells stops cell proliferation, and that this anomaly is overruled by SPINK2 coexpression. These experiments therefore clearly demonstrate that SPINK2 inhibits ACR and prevents the cellular stress initiated by unchallenged ACR expression. These results are discussed throughout the manuscript and presented in (a new) figure 9.

4. *The descriptive morphology of abnormalities of the Golgi and acrosome do not provide molecular mechanistic insight as to why/how the absence of SPINK2, an inhibitor acrosin, results in the observed phenotype.*

The experiment described in the previous section also permit to answer this question. We studied the impact of proacrosin expression on HEK293 cell proliferation and adherence by real time cell analysis (RTCA). We showed first that proacrosin can autoactivate (by Western blot) and induces a cellular stress leading to cell proliferation arrest and cell detachment (by RTCA), a phenotype similar to that observed in round spermatids from *Spink2* KO males. Moreover, Western blot results demonstrate that co-expression of SPINK2 with proacrosin prevents its auto-activation and rescues acrosin-dependent cell proliferation defects. These experiments demonstrate that proacrosin autoactivates within the cell and we therefore believe that SPINK2, which transits through the same cellular compartments, quenches this premature protease activity and prevents the described cascade of events leading to azoospermia.

5. *Fig. 1B should have age-matched fertile controls.*

Done, see new Fig 1B

6. *The hypothetical transcript in Fig. 3B should be confirmed by RT-PCR and sequence.*
See our response to point two. We now indicate that this hypothetical transcript is actually not transcribed.

Referee #2 (Remarks):

Minor comments:

- *At the end of results, it is written "SPINK2 haploinsufficiency is not be deleterious". Please, correct.*

This was corrected

- *At the beginning of the discussion, the sentence "chronic pancreatitis or Netherton syndrome" suggests that this syndrome corresponds to the pancreatitis when it's a dermatologic disease, as defined a few lines later. Please modify by "chronic pancreatitis and Netherton syndrome"*

This was corrected

- *In the next paragraph, acrosomal activity is released just before fertilization, not at fertilization.*

This was corrected

Two lines later, a space is missing before "among".

This was corrected

2nd Editorial Decision

10 April 2017

Thank you for the submission of your revised manuscript to EMBO Molecular Medicine. We have now received the enclosed report from the referee who was asked to re-assess it. As you will see the reviewer is now supportive and I am pleased to inform you that we will be able to accept your manuscript pending the following final editorial amendments:

1) Figures

- Your article currently has 10 figures and 4 EV figures. However, we feel like this could be shortened as follows: Figures 4 and 5 could be made into a single figure as they are described within the same paragraph in the results section, and the same goes for figures 7 and 8. We also feel that EV3 and EV4 could be put together within the same EV figure ( 8 figures and 3 EV figures). Further, as this would make 3 EV figures, 2 main figures could be moved to EV figures if possible. This would then result in 6 figures and 5 EV figures, shortening the length of the article, which is more attractive to the readers.

2) The author checklist:

- in section F-18, we do not fully agree with the written comment. You have genotyped the 2 brothers and the sequences should be supplied to EGA or a similar database. Please see <https://www.ebi.ac.uk/ega/home> for details

Please submit your revised manuscript within two weeks. I look forward to seeing a revised form of your manuscript as soon as possible.

***** Reviewer's comments *****

Referee #1 (Remarks):

The authors have addressed my concerns.

Corresponding Author Name: Pierre F. Ray

Journal Submitted to: EMBO MOLECULAR MEDICINE

Manuscript Number: EMM-2016-07461-V2